# Perfect flat band with chirality and charge ordering out of strong spin-orbit interaction

Hiroki Nakai[1✉] & Chisa Hotta [1✉]

Spin-orbit interaction has established itself as a key player in the emergent phenomena in modern condensed matter, including topological insulator, spin liquid and spin-dependent transports. However, its function is rather limited to adding topological nature to band kinetics, leaving behind the growing interest in the direct interplay with electron correlation. Here, we prove by our spinor line graph theory that a very strong spin-orbit interaction realized in $5d$ pyrochlore electronic systems generates multiply degenerate perfect flat bands. Unlike any of the previous flat bands, the electrons in this band localize in real space by destructively interfering with each other in a spin selective manner governed by the SU(2) gauge field. These electrons avoid the Coulomb interaction by self-organizing their localized wave functions, which may lead to a flat-band state with a stiff spin chirality. It also causes perfectly trimerized charge ordering, which may explain the recently discovered exotic low-temperature insulating phase of $CsW_2O_6$.

[1] Department of Basic Science, University of Tokyo, Meguro-Ku, Komaba 3-8-1, Tokyo 153-8902, Japan. ✉email: nakai-hiroki3510@g.ecc.u-tokyo.ac.jp; chisa@phys.c.u-tokyo.ac.jp

Electronic flat bands in momentum space are an ideal platform for achieving the highest correlation in a zero-bandwidth-limit[1–3]. A long history tells us that such flat bands naturally arise in a class of geometrically frustrated lattices like kagome, pyrochlore, and checkerboard lattices, which are well-understood based on the line graph theory and its analogs[4]. Recently, the importance of having flat bands in real correlated materials is highlighted in twisted bilayer graphene[5–7], where the relationship between superconductivity and magnetism has been extensively discussed. On top of finely tuning a magic 'twisting' angle, a flat band arises by structurally introducing a pseudo magnetic field onto a graphene layer[8].

There is another trend to add some topological nature to these flat bands[9–11], expecting emergent fractional quantum Hall states without a magnetic field, as they have a nonzero Chern number and mimic the Landau levels. A small spin–orbit coupling (SOC) helps to realize such nearly flat bands[12], which are experimentally observed in kagome lattice materials like CoSn[13] and twisted multilayer silicene[14]. Unfortunately, all these examples show that the perfect flatness of bands is sacrificed if the system gains topological properties[15].

Indeed, SOC rather enhances an itinerancy of electrons. Its major role had been to introduce some topological nature to the kinetic motion of particles. In SOC electronic systems[16–18], Berry phase is introduced to energy bands, which had been serving as a source of spin-dependent transports like anomalous Hall effect[19] and spin Hall effect[20,21]. A surface state of topological insulator[22,23] is a Dirac state, which is another distinguishing feature of energy bands induced by a weak SOC. When strong electronic interactions are present, the topological band insulator is transformed into a topological Mott insulator with a gapless surface spinon excitations[24]. In Kitaev materials[25], a very strong SOC creates a more exotic spin liquid phase[26] hosting Majorana fermions, and antiferromagnets with topological magnons[27,28]. Despite all these hallmark studies, there had been no example that the SOC gives an impact on the electronic correlation effect.

Here, we prove analytically that a SOC-induced spin-dependent hopping, which previously made the bands dispersive, perfectly flattens the energy bands of pyrochlore and kagome lattices when it becomes comparable to other transfer integrals. Most importantly, the SOC generates an SU(2) gauge field[29] and strictly selects the relative angles of electron spins. When electron wave functions have these spin angles, they destructively interefere[30] and localize in real space. We obtain an analytical form of such spin-twisted flat band wave function, allowing us to access the important but most unreachable physical regime, the strongest correlation. In analogy to the flat band ferromagnetism, the SOC flat band may select its form by polarizing its spins in a site-dependent manner avoiding the loss of on-site Coulomb energy, resulting in a stiff spin chirality. When the nearest neighbor Coulomb energy is introduced at quarter-filling, the wave function further optimizes its form to a trimerized shape by fully occupying half of the flat band wave functions, and becoming a spin-singlet state. This mechanism may explain the exotic trimerized charge ordering found in 5d pyrochlore $CsW_2O_6$[31], where one-quarter of the pyrochlore sites become perfectly vacant. The present model may provide a platform for testing the interplay of strong correlation and spin topology.

## Results

**Model system.** We introduce a minimal microscopic model for 5d pyrochlore oxides[32] (see Fig. 1a) with $CsW_2O_6$ as a specific example. A metallic $W^{5.5+}$ ion on a pyrochlore lattice is surrounded by a slightly distorted oxygen octahedron, and its electronic state is understood by considering the lowest Kramers

doublet of this ion ($E_2$ in Fig. 1b). The $E_2$ doublet comes out as the mixture of $t_{2g}$ triplet in a trigonal crystal field by introducing the strong SOC typical of the 5d electrons[33]. Its effective momentum deviates from the values of the regular octahedra, $J_{eff} = 3/2, J^z_{eff} = \pm 1/2$, by more than 10%. However, as in the case of Iridates, the $J_{eff}$-picture works well[34]. In the present quarter-filled case, the doublet carries 0.5 electrons on an average, where the energy levels are well separated as $E_1 - E_2 \sim 200$ meV (see Supplementary A and B). For such doublet described by a pseudo-spin, $\alpha = \uparrow, \downarrow$, a conventional Hubbard type of Hamiltonian[35,36] is written as a sum of hopping terms with spatially uniform transfer integral $t$ and Coulomb interaction $V$ between the nearest neighbor sites, $\langle i, j \rangle$, as well as the on-site $U$

$$
\begin{aligned}
\mathcal{H} &= \mathcal{H}_{kin} + \mathcal{H}_I, \\
\mathcal{H}_{kin} &= \sum_{\langle i,j \rangle} \sum_{\alpha, \beta} \left( -t\delta_{\alpha\beta} c^\dagger_{i\alpha} c_{j\beta} + i\lambda c^\dagger_{i\alpha} (\boldsymbol{v}_{ij} \cdot \boldsymbol{\sigma})_{\alpha\beta} c_{j\beta} \right) + \text{h.c.,} \\
\mathcal{H}_I &= \sum_j U n_{j\uparrow} n_{j\downarrow} + \sum_{\langle i,j \rangle} V n_i n_j,
\end{aligned} \tag{1}
$$

where $c_{j\alpha}$ annihilates an electron with pseudospin $\alpha$ at site-$j$, and $n_{j\alpha}$ and $n_j = n_{j\uparrow} + n_{j\downarrow}$ are their number operators. Eq. (1) has the same shape as an effective Hamiltonian for Iridates targeting $E_3$ doublet with $J_{eff} = 1/2$ and $J^z_{eff} = \pm 1/2$[36]. This is because both $E_3$ and $E_2$ consist of $a_{1g}$ and $e^\pi_g$ orbitals, and their difference appears only in the value of $\lambda/t$ (see Supplementary A Eq. (S8) and C). We note that due to small trigonal distortion, $t$ and $\lambda$ become slightly bond-dependent. For simplicity, we first approximate them as uniform and finally examine the effect of distortion. A bare atomic SOC which may amount to $\zeta \sim 200–300$ meV manifests as a spin-dependent hopping integral $\lambda$. A vector $\boldsymbol{v}_{ij}$ is a coefficient of Pauli matrices, $\boldsymbol{\sigma} = (\sigma_x, \sigma_y, \sigma_z)$, which is bond-dependent and is determined by the crystal symmetry. For a uniform pyrochlore lattice, we find $\boldsymbol{v}_{ij} = \sqrt{2} \frac{\boldsymbol{b}_{ij} \times \boldsymbol{d}_{ij}}{|\boldsymbol{b}_{ij} \times \boldsymbol{b}_{ij}|}$ with vectors $\boldsymbol{b}_{ij}$ and $\boldsymbol{d}_{ij}$ pointing from the center of the tetrahedron to the bond center and along the bond, respectively (see Fig. 1a). A mean-field phase diagram of a model similar to Eq. (1) is studied at half-filling for Ir-oxides[37] showing that a strong SOC generates a topological band insulator, a topological semimetal, and a topologically nontrivial Mott insulator in increasing $U$. There, an overall evolution of energy band structures in varying $\lambda/t$ and $U/t$ is studied in the context of finding a good Weyl point near the Fermi level[35,36]. In the present work, we notice that the SOC can drive another exotic phenomena, a perfect flat band and a trimerized charge ordering.

Let us first set $\mathcal{H}_I = 0$ and write down the energy bands by varying $\lambda/t$ in Fig. 1c. One finds a perfect flat band at the bottom when $\lambda/t = -2$. There is another case, $\lambda/t = 0$, with a flat band at the top, which is understood from the line graph theory. Introducing the SOC is known to destroy the perfectness of this top flat band[12] as one can see from the band structure for $\lambda/t = -0.5$. In the same context, it is shown that a perfect flat band cannot have a nonzero Chern number[15]. Notice that among the 32 bands, half contribute to the top flat band at $\lambda/t = 0$ which gradually gains a bandwidth by $\lambda < 0$, while at the same time the other dispersive half starts to shrink and finally becomes perfectly flat at $\lambda/t = -2$.

The flat bands at both $\lambda/t = 0$ and $-2$ touch the other dispersive bands at $\Gamma$-point. This band touching is neither an accidental degeneracy nor a typical symmetry-protected band degeneracy[38]. It is necessitated by the perfect flatness of bands, combined with some symmetry of the lattice[39,40]. When the perfect flatness of bands is lost at $\lambda < -2t$, the band touching disappears and a gap opens (see Supplementary D), and at half-filling, the system becomes a topological insulator.

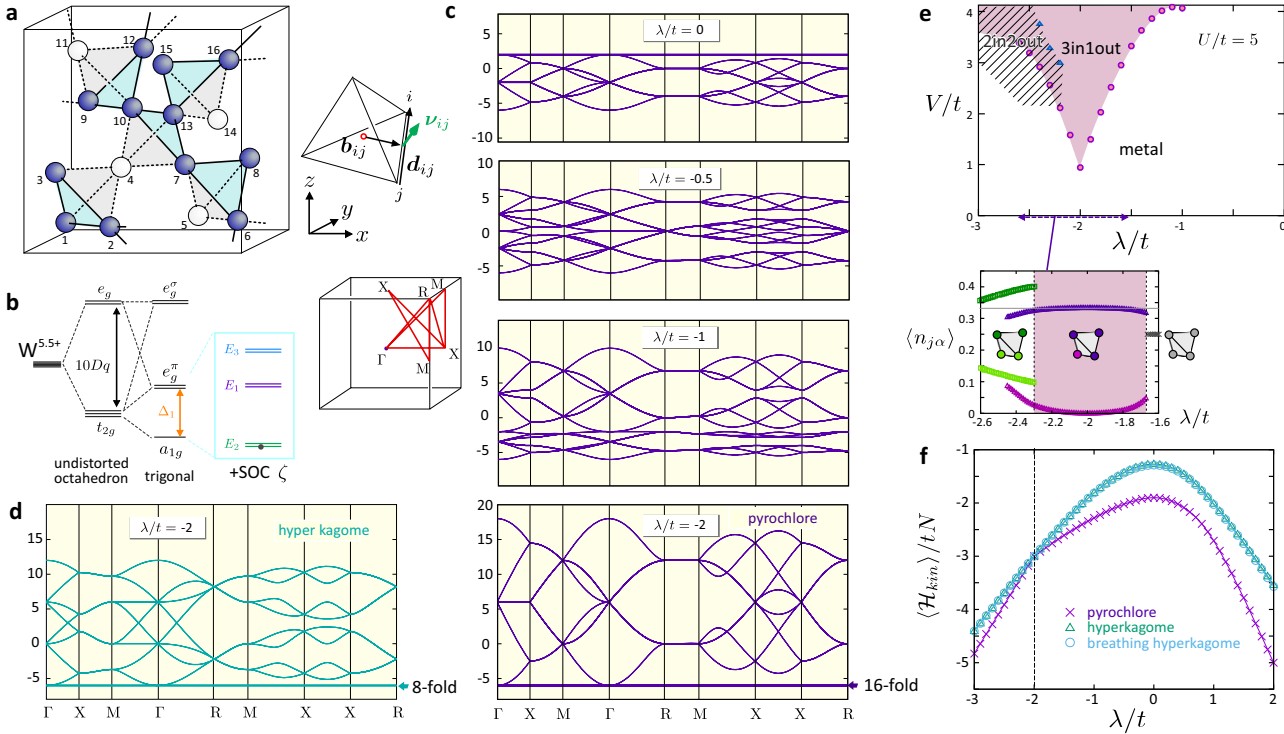

**Fig. 1 Nature of energy bands and the ground state of the pyrochlore electron systems with strong SOC. a** Unit cell of the pyrochlore/hyper-kagome lattice based on the W-ions, which includes four primitive pyrochlore cells. Filled and open circles represent the occupied and unoccupied sites in the trimerized charge order phase of $CsW_2O_6$, where the former forms a hyper-kagome lattice. $b_{ij}$ and $d_{ij}$ are vectors defined for bond $i$-$j$, determining the form of the SOC. **b** Energy scheme of single W-5$d$ surrounded by the oxygen ligand. **c** Noninteracting band structure (Eq. (1) with $U = V = 0$) of the pyrochlore lattice for $\lambda/t = 0, -0.5, -1, -2$. The $k$-paths are chosen as in the left side panel. **d** Noninteracting band structure of the hyper-kagome lattice obtained by depleting 1/4 of the lattice sites from the pyrochlore lattice with $\lambda/t = -2$. **e** Mean-field phase diagram of Eq. (1) on the plane of $\lambda/t$ and $V/t$ for $U/t = 5$. Circles/triangles represent the boundary where the energy of the 3-in-1-out becomes lower than the metallic state and 2-in-2-out state, respectively. (inset) Charge density $\langle n_{j\uparrow} \rangle = \langle n_{j\downarrow} \rangle$ of charge rich and poor sites at $V/t = 3$ of the mean-field solution. **f** Noninteracting band energy $\langle \mathcal{H}_{kin} \rangle$ of the pyrochlore, hyper-kagome, and the breathing hyper-kagome lattices.

We also show in Fig. 1d the band structure of a hyper-kagome lattice at the same $\lambda/t = -2$, obtained by depleting 1/4 of the pyrochlore sites, where we also find an 8-fold degenerate flat band at the same location.

**Phase diagram**. The SOC-induced flat band clarifies the origin of the trimerized charge ordering observed in $CsW_2O_6$[31]. Figure 1e shows a mean-field phase diagram at quarter-filling, corresponding to two electrons per tetrahedron. We approximate the Coulomb interaction terms $\mathcal{H}_I$ using a Hartree-type of mean-field and denote the solutions with $n$-charge-rich sites per tetrahedron as $n$-in-$(4 - n)$-out. A trivial paramagnetic metallic state with uniform charge and spin distribution are dominant when the Coulomb interaction is small. There is an emergent 3-in-1-out state extending at around $\lambda/t = -2$, which has 2/3 electrons per hyper-kagome site, keeping 1/4 of the site almost perfectly empty (see the inset of Fig. 1e). The 2-in-2-out phase with about 0.35:0.15 charge disproportionation is stabilized only at $\lambda/t \lesssim -2$.

The reason why 3-in-1-out is stable is understood by comparing the band energies $\langle \mathcal{H}_{kin} \rangle$ in Fig. 1f when pyrochlore and hyper-kagome lattices host $8N_c$ electrons, where $N_c$ is the number of unit cells. A band-energy-gain is always larger for a pyrochlore lattice with a larger coordination number and thus having the larger bandwidth. Indeed, the $\lambda > 0$ region of the phase diagram is dominated by a trivial metallic phase even for large $U$ and $V$. However, at $\lambda/t = -2$, the pyrochlore and hyper-kagome band energies become degenerate because all the electrons fill the bottom flat bands for both cases. The mean-field interaction

energy is roughly evaluated by hand as $E_I^{metal} = U + 12V$ and $E_I^{3i1o} = 4U/3 + 32V/3$ per unit cell for metal and 3-in-1-out, respectively, which is consistent with our numerical evaluation based on a mean-field approximation(see Supplementary E). Then, the introduction of $V/t \gtrsim 1$ stabilizes the 3-in-1-out state against the metallic phase.

**Spinor line graph theory**. The perfect flat band at $\lambda/t = -2$ cannot be explained within any of the previous frameworks. Here, we develop a spinor line graph theory to prove the existence of SOC-induced flat bands, which can be applied to general line-graph-related lattices. To this end, we first overview the flat band theory for line graphs. Figure 2a, b shows the relationships between the original lattice and its dual lattice described by red circles. The pyrochlore lattice is a line graph of its dual lattice, a diamond lattice, and by connecting pyrochlore and diamond sites and deleting pyrochlore bonds, one reaches a bipartite graph with blue bonds. The same relationship holds between the kagome–honeycomb lattices.

Let us introduce an incidence matrix of a graph theory, $T_{OD}$, to describe the relationship between the original lattice and its dual lattice. It is an $N \times N_D$ matrix and has one row for each pyrochlore site and one column for each diamond site, where $N = 16N_c$ and $N_D = 8N_c$ denote the number of pyrochlore and diamond lattice sites, respectively. The entry in row-$i$ and column-$m$ is 1 if pyrochlore-site-$i$ and diamond-site-$C_m$ are connected by a blue bond. If we take a product of the incidence matrix with its transpose matrix $T_{DO} = {}^{t*}T_{OD}$ as $(T_{OD}T_{DO})$, its $ij$-

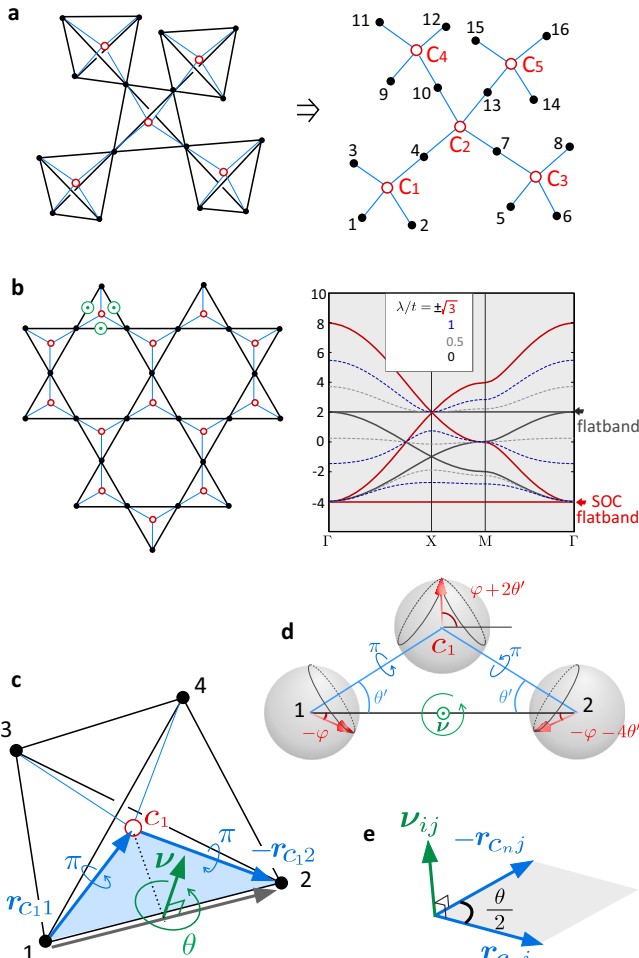

**Fig. 2 Spinor line graph theory. a** Pyrochlore lattice and its dual diamond lattice ($C_j$) in the red circle. The two lattices are connected and form a bipartite graph on the right panel. **b** Kagome and its dual honeycomb lattice, where $\boldsymbol{\nu}_{ij} = (0, 0, \pm 1)$. Noninteracting band structures for $\lambda/t = 0$ to $\sqrt{3}$ are shown. **c** Unit tetrahedron. Hopping $1 \to 2$ rotates the spin orientation by $\theta$ about the $\boldsymbol{\nu}_{21}$-axis, whereas hopping through $1 \to C_1 \to 2$ rotates the spin orientation twice by $\pi$ each about the $\boldsymbol{r}_{C_1 1} = (1, -1, 1)$ and $\boldsymbol{r}_{C_1 2} = (-1, 1, 1)$ axes for site-1 and 2 in Fig. 1a. Here, $|r_{C_1 1}|$ is 8 times larger than the true $1 \to C_1$ vector when taking the length of the cell as unity. **d** Example of spin rotation along two paths that fulfill Eq. (5). **e** Condition to have SOC flat band in Eq. (6).

entry becomes 1 when there is a connection between $i$th and $j$th pyrochlore sites mediated via the diamond site through two blue bonds. The diagonal element of ($T_{OD}T_{DO}$) has entry-2 since each pyrochlore site can be transferred to its two neighboring diamond sites and come back. Using this product form, a matrix representation of a tight-binding Hamiltonian of the pyrochlore lattice is written as

$$\hat{H}_{\text{pyrochlore}}(\lambda = 0) = 2t\hat{I} - tT_{OD}T_{DO}, \qquad (2)$$

where $\hat{I}$ is a unit matrix. According to this equation, if there is an $N$-dimensional vector $\boldsymbol{\varphi}_l$ that fulfills $T_{DO}\boldsymbol{\varphi}_l = 0$, it also satisfies $\hat{H}_{\text{pyrochlore}}\boldsymbol{\varphi}_l = 2t\boldsymbol{\varphi}_l$. A set of such vector forms a kernel (null-space) $\{\boldsymbol{\varphi}_l\}$ of $T_{DO}$. Since $T_{DO}$ is non-square, the number of independent $\boldsymbol{\varphi}_l$, namely the dimension of the kernel is at least $N - N_D = 8N_c(> 0)$. It means that there exist at least $(N - N_D)/N_c = 8$ flat bands in the pyrochlore lattice with an energy $2t$, which is the one found in Fig. 1c at $\lambda/t = 0$, where considering the spin degeneracy, the number of flat bands is doubled.

The extension of the line graph theory to $\lambda \neq 0$ is not straightforward, since the hopping term is rewritten as

$$\mathcal{H}_{\text{kin}} = \sum_{\langle i, j \rangle} -\sqrt{t^2 + 2\lambda^2}\, \boldsymbol{c}_i^\dagger U_{ij} \boldsymbol{c}_j, \quad U_{ij} = e^{-i\frac{\theta}{2}\hat{\boldsymbol{\nu}}_{ij}\cdot\boldsymbol{\sigma}}, \qquad (3)$$

and includes a non-Abelian SU(2) gauge field $U_{ij}$[29], where $\boldsymbol{c}_j^\dagger = (c_{j\uparrow}^\dagger, c_{j\downarrow}^\dagger)$ and $\hat{\boldsymbol{v}} = \boldsymbol{v}/|\boldsymbol{v}|$ is a unit vector. The gauge field along $j \to i$ enforces an SU(2) spin rotation about the $\boldsymbol{v}_{ij}$-axis by an angle $\theta = 2\arctan(\sqrt{2}\lambda/t)$. We want to construct another incidence matrix $\tilde{T}_{OD}$, whose $jn$-entry represents a spin-rotating hopping of an electron from the $j$th pyrochlore site to the $C_n$th diamond site. It should be such that the $ij$-entry of ($\tilde{T}_{OD}\tilde{T}_{DO}$) will reproduce the complex hopping of Eq. (3). In hopping twice along the blue bonds, electron spin is rotated twice, ending up with the same state as rotated by $\theta$ about the $v$-axis. As we show in Fig. 2c, considering the symmetry of the tetrahedron, the rotation axis in hopping $1 \to C_1$ is uniquely chosen along the bond pointing from the vertex to the center of the tetrahedron, which we denote as $\boldsymbol{r}_{C_1 1}$. The rotation angle is also uniquely chosen as $\pi$. Resultantly, an incidence matrix $\tilde{T}_{OD}$ including the effect of SU(2) gauge field for $\lambda \neq 0$ is given as

$$(\tilde{T}_{OD})_{jC_n} = \begin{cases} -i(\boldsymbol{r}_{jC_n} \cdot \boldsymbol{\sigma}) = |\boldsymbol{r}_{jC_n}|e^{-i\frac{\pi}{2}\hat{\boldsymbol{r}}_{jC_n}\cdot\boldsymbol{\sigma}} & \text{(connected)} \\ 0 & \text{(otherwise)} \end{cases} \qquad (4)$$

As shown in the caption of Fig. 2, we take $|\boldsymbol{r}_{jC_n}| = \sqrt{3}$ for convenience, while this value only influences the coefficient of the second term of Eq. (5). Since the spin degrees of freedom is explicitly included, the matrix has twice as large dimension as $T_{OD}$, and fulfills $\tilde{T}_{DO} = {}^{t*}\tilde{T}_{OD}$.

In the similar manner as Eq. (2), the incidence matrix is related to a hopping matrix $\hat{H}_{\text{pyrochlore}}$, i.e., a real-space matrix representation of Eq. (3), as

$$\hat{H}_{\text{pyrochlore}}(\lambda/t = -2) = -6t\hat{I} + t\tilde{T}_{OD}\tilde{T}_{DO}, \qquad (5)$$

when and only when $\lambda/t = -2$. To understand why $\lambda/t$ needs to take this value, we show in Fig. 2d an example; consider a spin at site-1 pointing inside the $1 - C_1 - 2$ triangular plane with angle $-\varphi$. For the present geometry of the pyrochlore lattice, we have an angle $\theta' = \arccos(\sqrt{2/3})$ spanned by $1 \to 2$ and $1 \to C_1$. When the spin is transferred by ($\tilde{T}_{OD}\tilde{T}_{DO}$) it rotates by $\pi$ twice, takes the angle $(\varphi + 2\theta')$ at site-$C_1$ and points to $(-\varphi - 4\theta')$ at site-2. When $\theta' = -\theta/4$, this operation replaces the $\theta$-rotation about the $v$-axis. This geometrical condition gives $\lambda/t = -2$, and is a unique solution to fulfill Eq. (5). A kernel of $\tilde{T}_{DO}$ is a manifold of eigenstate of $\hat{H}_{\text{pyrochlore}}(\lambda/t = -2)$ with a constant energy $-6t$, and has a dimension $2(N - N_D)$. Therefore, we find $2(N - N_D)/N_c = 16$ flat bands at the energy bottom $-6t$.

A guide to design such SOC flat band is simple. The above mentioned geometrical condition for angle $\theta$ can be generalized to

$$\frac{\boldsymbol{r}_{iC_n} \times (-\boldsymbol{r}_{jC_n})}{\boldsymbol{r}_{iC_n} \cdot (-\boldsymbol{r}_{jC_n})} = -\tan\frac{\theta}{2}\hat{\boldsymbol{v}}_{ij} = -\frac{\lambda}{t}\boldsymbol{v}_{ij}, \qquad (6)$$

which is schematically shown in Fig. 2e. Using Eq. (6), one may search for a lattice geometry that gives a reasonable value of $\lambda/t$. Another expression for this condition uses a Wilson loop operator $\mathcal{A}_{jC_n i}$ around the closed loop $i \to C_n \to j$. Eq. (6) is equivalent to having $\mathcal{A}_{2C_1 1} = U_{12}U_{2C_1}U_{C_1 1} = e^{-i\frac{\theta}{2}\hat{\boldsymbol{v}}_{12}\cdot\boldsymbol{\sigma}}e^{-i\frac{\pi}{2}\hat{\boldsymbol{r}}_{2C_1}\cdot\boldsymbol{\sigma}}e^{-i\frac{\pi}{2}\hat{\boldsymbol{r}}_{C_1 1}\cdot\boldsymbol{\sigma}} = -I$. The condition means that for two dimensional lattices, the SOC vector $\boldsymbol{v}$ shall point in the out-of-plane direction, and also when $\theta = \pi$, the $ij$-bond takes $t = 0$ and $\lambda \neq 0$. Since such parameter

values may not easily be realized, the edge-shared lattices like square, checkerboard, and honeycomb lattices may not be considered as realistic examples.

The spinor line graph theory is applied to kagome and hyper-kagome lattices. For a kagome lattice, as shown in Fig. 2c, a usual $\lambda = 0$-flat band at $+2t$ starts to gain bandwidth with $\lambda \neq 0$, while a dispersive bottom band shrinks and becomes a SOC flat band at $-4t$ when $\lambda = \pm\sqrt{3}$. (See Supplementary F for details).

**Destructive interference**. Although treating a quantum many-body model beyond a mean-field level is too challenging in general, our case with a zero-bandwidth at $\lambda = -2t$ may become simpler since it practically corresponds to a strong coupling limit which can be partially treated analytically. Among the one-body flat bands orbitals, $\varphi_{l\alpha}$ ($l = 1, \cdots 16N_c$, $\alpha = \uparrow, \downarrow$), half are filled when we consider $CsW_2O_6$. The set of one-body flat bands is chosen as their linear combinations, such that they minimize the interaction energy loss in total when they are combined to form a many-body flat band wave function.

The $m$th one-body flat-band eigenstate of $\mathcal{H}_{kin}$ including SOC is written as $|\psi_m\rangle = \sum_{j,\alpha} \varphi_{j\alpha}^m c_{j\alpha}^\dagger |0\rangle$, where the complex coefficients $\varphi_{j\alpha}^m$ are the elements of $32N_c$-dimensional vector $\tilde{\varphi}_m$ that fulfills $\tilde{T}_{DO}\tilde{\varphi}_m = 0$. This condition is factorized to the condition for each tetrahedron; it prohibits a net propagation of electrons from four pyrochlore sites labeled by $j \in n$ to an $n$th diamond site as

$$\sum_{j \in n} (-i\mathbf{r}_{C_nj} \cdot \boldsymbol{\sigma}) \begin{pmatrix} \varphi_{j\uparrow}^m \\ \varphi_{j\downarrow}^m \end{pmatrix} = 0, \qquad (7)$$

which should be fulfilled for all tetrahedra $n = 1, \cdots, N_D$. In visualizing this equation, we first set a fictitious SU(2) spinor $\chi_n$ (two-dimensional vector) at the $n$-th tetrahedron center pointing somewhere as in Fig. 3a. Suppose that the spins on four pyrochlore sites point in the directions rotated by $\pi$ from this spinor about the blue-bonds. Among these four spins, if some have finite weight $\varphi_{j\alpha}^m$ in the wave function, they need to be canceled out by Eq. (7).

When considering the two adjacent tetrahedra, a spin shared by them should fulfill the two conditions. This spin shares the same $\pi$-rotation axis in hopping to the diamond sites on both sides. Therefore, if it has a finite population in the wave function, the two fictitious spinors on both sides are enforced to point in the same direction. One example of $|\psi_m\rangle$ is given as such that they form a closed loop consisting of an even number of bonds, shown in Fig. 3b. By assigning $+1$ and $-1$ weights alternatively along the loop while fixing their spin direction in a way mentioned above, a single electron is perfectly localized on the loop. This is because if it wants to hop outside the loop, its weights are canceled out by Eq. (7), which is the physical meaning of a destructive interference[30] or a kinetic frustration. The product of one-body flat band wave functions becomes an eigenstate of $\mathcal{H}_{kin}$, which is also an eigenstate of $\mathcal{H}_I$, namely of the whole Hamiltonian.

We now consider a trimerized charge-ordered state based on a flat band wave function. There are $(16N_c + 2)$ linearly independent one-body states that belong to the pyrochlore flat band including the pseudo-up/down-spins and band touching ones. Among them, one can choose $4N_c \times 2$-independent ones, forming a loop consisting of ten sites that belong to the hyper-kagome lattice which we call loop-10 as shown in Fig. 3c (see Supplementary G). A 3-in-1-out many body flat band wave function is thus given in a factorized form, $|\Psi_{3in1out}\rangle \propto \prod_{n,\sigma} \hat{\psi}_{n,\sigma}^{10}|0\rangle$, using a single electron operator of loop-10, where $|\psi_n^{10}\rangle = \hat{\psi}_{n\sigma}^{10}|0\rangle$. The index $\sigma = \uparrow, \downarrow$ of $\hat{\psi}_{n,\sigma}^{10}$ corresponds to $\chi_n = (1,0)$ and $(0,1)$. In the present quarter-

filled case, since we need to put two electrons per tetrahedron, namely $8N_c$ electrons on $4N_c \times 2$-independent loop-10 states, they accomodate both pseudo-up and down spins and are fully occupied. Therefore, $|\Psi_{3in1out}\rangle$ is a nonmagnetic singlet state. These loop-10's have finite overlap and distribute uniformly over the whole hyper-kagome lattice with all sites having the same electron occupancy of 2/3.

Apart from the case of $CsW_2O_6$, there is purely theoretical interest in lower fillings. For no more than half-filling of flat bands, one can prepare a many-body wave function consisting of a product of loops, e.g., loop-6 state written in Fig. 3b that fulfill Eq. (7). Here, by selecting the spin orientation for each, the whole wave function is constructed as such that it gives the lowest $\langle \mathcal{H}_I \rangle$. When all these constituent one-body functions have finite overlap with some others and cannot be disconnected into two groups, one can fully avoid the double occupancy of electrons on all sites by polarizing $\chi_n$ for all $n$ in the same direction, which gives $\langle U n_{i\uparrow} n_{i\downarrow} \rangle = 0$. When $V = 0$, this wave function becomes the exact and unique ground state of the Hamiltonian. This context is analogous to the mechanism of flat band ferromagnetism of a Hubbard model[1,2]; electrons choose which of the localized one-body flat-band wave functions to occupy by fully polarizing their spins at finite-$U$ since Pauli's principle helps the electrons to avoid double occupancy in space.

When $\chi_n$ for all tetrahedra point in the same direction, the many-body flat band state exactly keeps the relative angles of the spins on four sublattices, which indicates the stiff chiral ordering. As shown in Fig. 3d there are eight species of triangles in a unit cell, whose spin orientations are shown for the case where the fictitious spinor points in the $+z$-direction. These pseudo-spins are exposed to an internal magnetic field generated by an SU(2) gauge field, and its flux equals half of the solid angle $\Omega_{ijk}$ subtended by the spin directions around the triangle. We evaluated $\Omega_{ijk} = \mathbf{n}_i \cdot (\mathbf{n}_j \times \mathbf{n}_k)$ for four independent triangles in Fig. 3d as a function of angle $\Theta$ of the fictitious spinor about the $+z$-axis. We define a unit vector $\mathbf{n}_j$ parallel to the pseudo spins with the right-hand rule about $+z$-axis. At $\Theta = 0, \pi$ we find maximum amplitude, $\Omega_{321} = \pm 16/27$. In this case, this scalar chirality contributes to a $xy$-component of an anomalous thermal Hall conductivity for insulators or it might affect $\sigma_{xy}$ for metals[41,42].

## Discussion

Concerning the experimental findings, an important question is whether the actual material parameters really fit to our scenario. It is known that the $5d$ electrons are more extended in space with a reduced value of on-site Coulomb repulsion $U \sim 1 - 2$ eV[5] and an enhanced bandwidth, which may favor a metallic state[32,34]. However, a large atomic SOC, $\zeta$, comparable to transfer integral($t$) usually dominates the $t_{2g}$ orbitals and splits them into higher $J_{eff} = 1/2$ doublet and lower 3/2 quartet. A Mott insulating $Sr_2IrO_4$ is reported to have $t \sim 0.3$ eV and $\zeta \sim 0.5$ eV[43], and parameters of a honeycomb Kitaev material $Na_2IrO_3$ are evaluated as $t \sim 0.27$ eV and $\zeta \sim 0.39$ eV from the first principles calculation[44]. In $CsW_2O_6$, the value of SOC should be $\zeta \sim 200 - 300$ meV, which is considered to be about half of that of $5d$ Iridates. A trigonal distortion of the crystal further splits the $J_{eff} = 3/2$ quartet into two, and the lowest $E_2$ doublet with $J_{eff}^z \sim \pm 1/2$ and $J_{eff} \sim 3/2$ is focused(see Fig. 1b).

In $CsW_2O_6$ the distortion angle, $\alpha = 55.71°$, is slightly larger than the regular octahedron $54.74°$. Based on this information, we examined in detail the energy-level splitting of W-$5d$ in a trigonal crystal field in Supplementary A and B, and by associating the results with the energy band structure of the first principles calculations without SOC, we estimated a set of material parameters

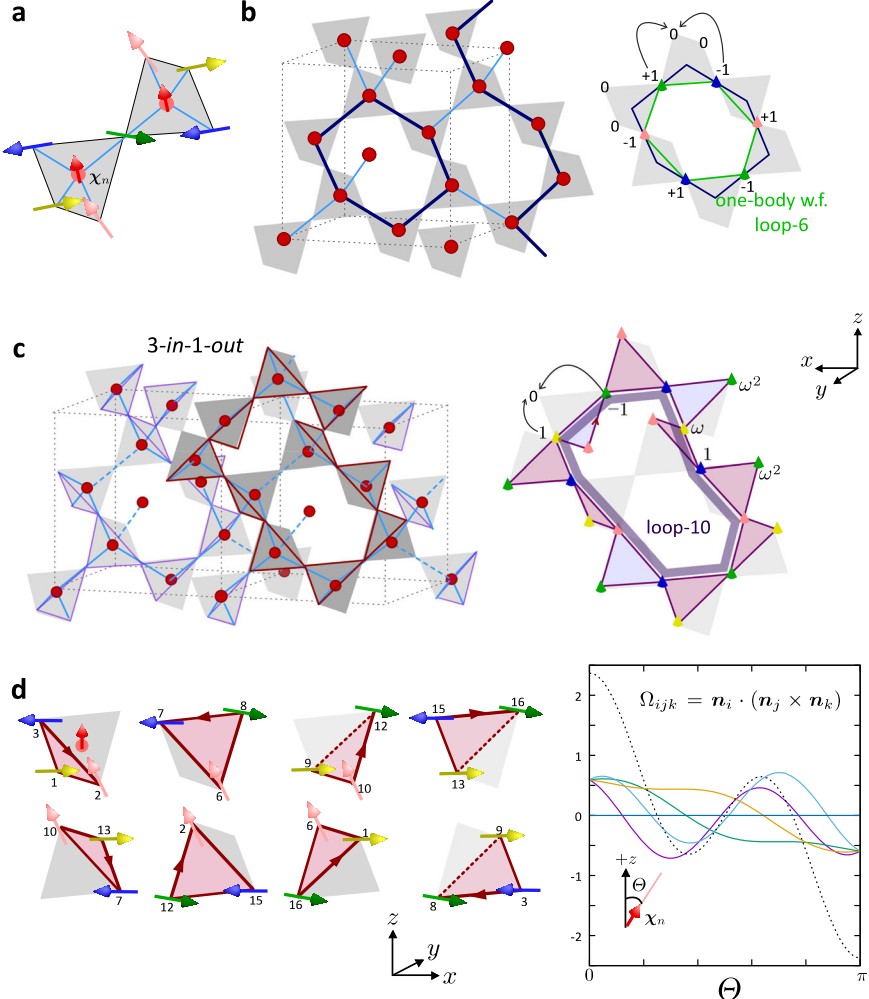

**Fig. 3 Spin-dependent real space configurations of the SOC flat band states. a** Relative spin configuration of flat band states. Spins on the vertices of the tetrahedra are created from the center spin(red arrow) via SU(2)-rotation by $\pi$ about the axes pointing from the center toward the vertex. The orientation of fictitious spins at the center of two adjacent tetrahedra, $C_i$ and $C_j$, point in the same direction as far as they are connected by a finite population of spins at the vertex between $C_i$ and $C_j$. **b** Schematic illustration of a one-body flat band wave function of loop-6. Weights of electrons on these sites align in a staggered manner $+1, -1, \cdots$, and spins are oriented in different directions marked with different colors, which are relatively fixed. **c** Schematic illustration of a loop-10 one-body flat band wave function on a hyper-kagome lattice. The product of these loop-10 gives the 3-in-1-out state. **d** Four different directions of spins in different colors, which form eight different types of triangles in the many-body flat band wave function. The solid angle is evaluated for four pairs of triangles separately as a function of angle $\Theta$ about a $+z$-direction, and its summation given in broken line takes the maximum amplitude for $\Theta = 0, \pi$.

as $t \sim 0.06$ eV, $10Dq \sim 2$ eV, and $\Delta_1 \sim 0.23$ eV. By introducing $\zeta \sim 0.1 - 0.15(10Dq)$, the energy levels of the three doublets are obtained and we find $E_1 - E_2 \sim 0.1(10Dq) = 0.2$ eV, which is reasonably large to justify our approximation dealing with only $E_2$ doublets.

At $\zeta = 0$ and in a trigonal crystal field, the $E_2$ doublet has a character of $a_{1g}$, while with increasing $\zeta$ the contribution from $e_g^\pi$ levels becomes the same order as $a_{1g}$. The spin-dependent hopping integral $\lambda$ originates from the direct and oxygen-mediated indirect hoppings between $e_g^\pi$ and $a_{1g}$, and has different signs from $t$ coming from the $a_{1g}$–$a_{1g}$ and $e_g^\pi$–$e_g^\pi$ hoppings. We made a microscopic evaluation of $\lambda/t$ of $CsW_2O_6$ based on the Slater-Koster parameter and found that the ratio ranges between $\lambda/t \sim -3$ to $-1$ depending on the ratio of direct hopping against indirect hopping(see Supplementary C). Our SOC-induced flat band can thus be reasonably realized in the material. We also notice that in our theory, one does not need strictly $\lambda/t = -2$ to have a trimerization, as the phase diagram shows that there is some sort of pinning effect to the flat bands when the electronic interactions are finite.

In the $J_{\text{eff}}$–picture the $t_{2g}$ orbital momentum $L^{\text{eff}} = 1$ resembles the $p$-orbital representation with its sign taken as minus, where we find $J_{\text{eff}} = -L^{\text{eff}} + S$ as good quantum numbers[45]. Then, the magnetic moment $M = 2S - L^{\text{eff}}$ becomes zero for the undistorted octahedron, while for the present case the admixture of levels coming from small trigonal distortion gives finite moment $\langle M_z \rangle$ still about half of that of the full moment of the electron, while it is difficult to compare this directly with the available experimental results.

In the low-temperature phase II, we expect the trimerized flat band state, which has a Mott gap. This explains the sharp increase of the resistivity at the transition temperature[31]. The many-body flat band state on a hyper-kagome lattice we obtained is non-magnetic, which may explain a finite spin gap.

Before the recent discovery of trimerized charge ordering that keeps the Anderson condition[31], $CsW_2O_6$ was considered to undergo a Peierls-type of metal-insulator transition[46]. This was partially because the DFT calculation showed a large enhancement of the density of states near the Fermi level[47], which was ascribed to the electronically driven structural-metal-insulator

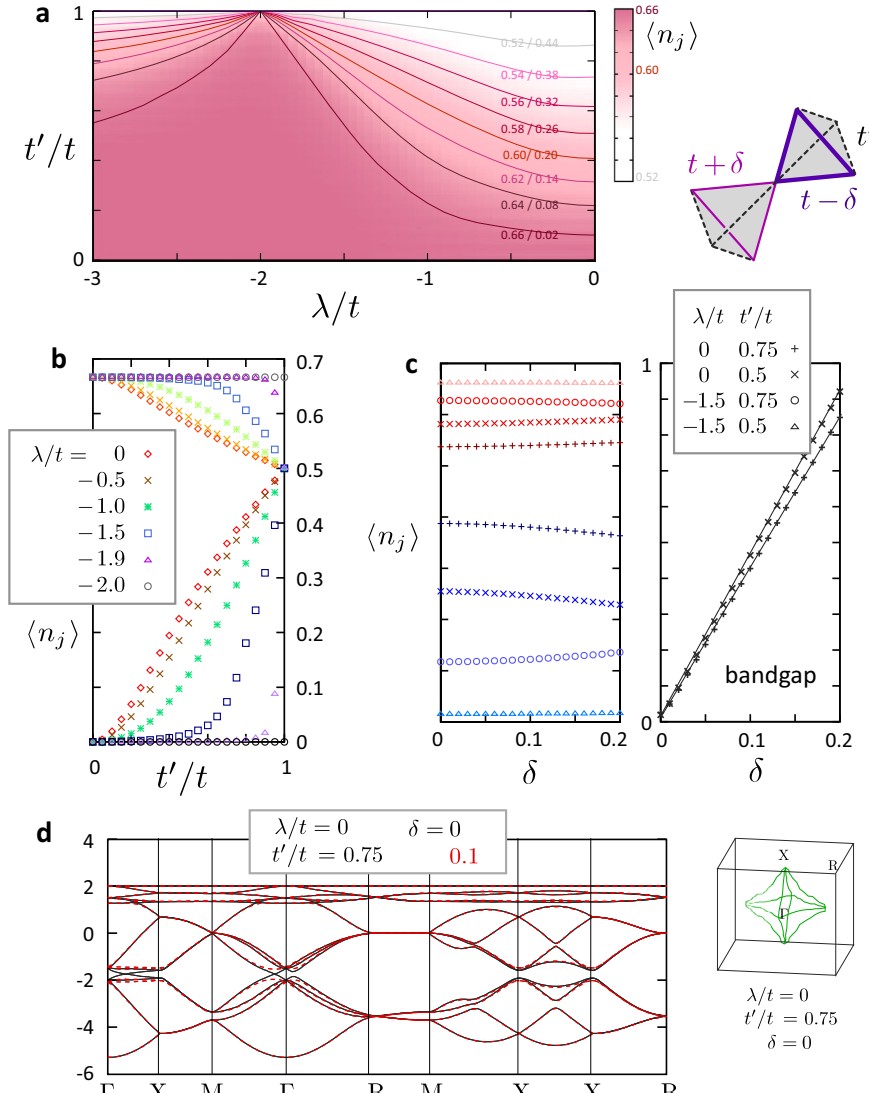

**Fig. 4 Examination of the effect of lattice distortions: $U = V = 0$ and the transfer integrals being modified for $t \to t'$, $t \pm \delta$. a** Charge density $\langle n_j \rangle = \langle n_{j\uparrow} + n_{j\downarrow} \rangle$ for $\delta = 0$ on the plane of $\lambda/t$ and $t'/t$. Contour lines with charge rich/poor densities are drawn. **b, c** $\langle n_j \rangle$ as function of $t'/t$ and $\delta$. Right panel is the band gap for $\delta \geq 0$ when $t'/t = 0.75$ and 0.5. **d** Band structures for $\lambda = 0$ with $t'/t = 0.75$ comparing the cases with $\delta = 0$ (solid) and 0.1 (broken line). Right panel is the Fermi surface for the $\delta = 0$ case.

transition to a zig-zag-like one-dimensional structure. Other first-principles calculations supported this picture arguing that the SOC enhances the nesting instability[48]. Also, a certain amount of lattice distortion takes place at the transition, and a hyper-kagome lattice based on the charge-rich sites shows breathing into large and small triangles with the difference in their bond length by 2%[31], which seemingly supports the Peierls transition.

To clarify that the SOC is the driving force of the trimerized charge ordering, we finally show that it is difficult to attain such perfect charge disproportionation solely by the lattice distortion and without $\lambda$. Considering the type of structural distortion taking place in the material, we modify the originally uniform $t$ to three classes: $t'$ shown in broken lines that connect the charge-rich and poor sites, and $t \pm \delta$ which form small/large triangles of a hyper-kagome lattice. Figure 4a shows the density plot of charges on the plane of $\lambda$ and $t'$ for $\delta = 0$. Only near $\lambda \sim -2t$, one can attain a nearly perfect (2/3 : 0)-ratio of charge disproportionation at $t' \lesssim t$. Notice that in general, $t'$ can never be smaller than even half of $t$ with such lattice distortion, although we examined the whole range of $t'/t = 0$ to 1. Figure 4b, c is the variation of rich/

poor $\langle n_i \rangle$ as functions of $t'$ and $\delta$, and a bandgap at the Fermi level. There are two notable features. The charge density can be very close to the flat band ones even though $\lambda$ is off $-2t$, once we decrease $t'$ slightly from 1. In contrast, the breathing effect, $\delta$, typical of the "Peierls transition", does not change the charge density, even when the bandgap increases as we see for the case of $\lambda = 0$; the gap opening at $\delta = 0.1$ with the disappearance of the Fermi surface on the left panel is shown in Fig. 4d.

In revisiting the aforementioned previous works, the enhanced density of states does not mean the Peierls instability but may rather fit the scenario of possible SOC induced flat band, which may not be perfect, but would be enough to drive the system to a trimerized charge ordering. According to our theory, this charge order is different from the conventional ones driven mostly by the Coulomb interaction $V$. The interplay of SOC and transfer integral is its main source. $U$ and $V$ only indirectly support it, since the flat-band wave function has an advantage over trivial electronic states in that, they could self-organize their shape freely within the manifold of flat-band eigenstates and optimize their charge configuration to avoid the Coulomb interactions.

The present picture might be examined by an anomalous thermal Hall measurement in the insulating phase or an anomalous Hall electronic transport in the metallic state by the hole-doping to the material. In the previously known cases of the intrinsic anomalous Hall effect, often the SOC acting on the conducting electrons[42] or the localized moments working as spatially coplanar internal field onto the conducting electrons[41] was considered as a source of the emergent gauge field. In our case, the SOC is playing a more crucial role, as it works to kill their momentum $k$ and strictly selects the orientation of pseudo-spin moments. These electrons may virtually propagate in space since it is on a flat band. It is thus beyond the scope of the present transport theories on how such features may appear in the transport phenomena.

## Data availability
The data that support the findings of this study are available on request from the authors.

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

## Acknowledgements

We appreciate Youichi Yamakawa for useful information on the first principles band structure. We thank Taka-hisa Arima, Yoshihiko Okamoto, and Masataka Kawano for discussions. The work is supported by JSPS KAKENHI Grants Nos. JP17K05533, JP18H01173, JP21H05191, and JP21K03440 from the Ministry of Education, Science, Sports, and Culture of Japan.

## Author contributions

C.H. designed the project and wrote the paper. H.N. constructed the model and performed the mean-field calculation. Both constructed the theory together and equally contributed to this work.

## Competing interests

The authors declare no competing interests.
