## [Peer Review File · Nature Communications]

REVIEWER COMMENTS

Reviewer #1 (Remarks to the Author):

Report of Manuscript#: NCOMMS-21-13841-T

In the manuscript "Perfect flat band and chiral-charge ordering out of strong spin-orbit interaction", H. Nakai et al. developed a spinor line-graph theory and proposed that perfect flat bands could exist in the presence of strong spin-orbit coupling. Combined with the mean-field theory, they explained the charge ordering found in the pyrochlore material CsW₂O₆. The perfect spinful flat bands in line-graph lattices and the spinor line-graph theory is very interesting and a timely topic.

Here are a couple of questions or comments:

1. In Eq. (2), the tight-binding Hamiltonian of a bipartite graph should have a chiral symmetry which leads to the flat bands being localized on the zero energy and connected with the other dispersive bands below and above. It means that there must have interference between the pyrochlore and diamond sublattices. Thus, Eq. (3) and (6) cannot be obtained directly from Eq. (2). Is there any approximation? We suggest the authors provide more details about the derivation.
2. As studied in Ref.[12], for an s-orbital Kagome lattice, the presence of spin-orbit coupling opens a gap between the flat bands and the dispersive bands at Γ point and makes the flat bands be topological. While in FIG. 1c and FIG. 1d (with $\lambda=-2t$), the perfect flat bands seem to degenerate with the dispersive bands at the Γ point. Is the degeneracy accidental or protected by any symmetry? What if λ slightly off the fine-tuning value $-2t$? I would also be interested in the topology of the flatten bands if there opens a gap at the Γ point.

Overall, this work is of interest to me and I would like to recommend the publication of this manuscript in Nature communication after the above questions are properly addressed.

Reviewer #2 (Remarks to the Author):

Authors present a theoretical analysis of the origin of a flat band which appears for the pyrochlore lattice in the limit of strong spin-orbit interaction (SOC). For this purpose they extend the line graph theory to the case of spin-dependent hopping. The model is applied to CsW₂O₆ which undergoes a structural transition to a trimerized phase.

To my opinion, the description of the spinor line graph theory is too technical and is more suitable for a specialized journal.

The explanation of the origin the trimerization in CsW₂O₆ can be interesting for a broad solid-state community. However, coming from a band structure community I am confused by the model suggested in the paper and, especially, by its relevance for CsW₂O₆. I would be glad if Authors could clarify the points mentioned below. Before that I cannot recommend the paper for publication.

1. On p.1 Authors state that the E2 doublet carries momentum $J_{\text{eff}}=3/2$, $J_{\text{eff}}^z=\pm 1/2$. However, trigonal crystal field (CF) couples $J_{\text{eff}}^z=\pm 1/2$ states with $J_{\text{eff}}=1/2$ and $3/2$ and, consequently, E2 cannot be an eigenstate of J_{eff} .

2. What approximations are made when deriving the Hamiltonian (1)? In particular:

Without SOC the E2 doublet is formed by pure a_{1g} states while with increasing SOC strength it becomes a mixture of a_{1g} and $e_g\pi$ states. On the pyrochlore lattice the magnitudes of a_{1g} - a_{1g} and $e_g\pi$ - $e_g\pi$ hopping matrix elements are very different. Is it taken into account in (1)?

3. Bands in the $\lambda/t=0$ panel in Fig. 1b look like bands formed by s -like states put on the pyrochlore lattice which is OK for a_{1g} bands. On the other hand, the $\lambda/t=-2$ bands look very similar to $J_{\text{eff}}=1/2$ bands which I get from band structure calculations for 5d pyrochlore compounds in the limit of large SOC. Naively, I would expect different dispersion for bands formed by a linear combination of $J_{\text{eff}}=1/2$ and $J_{\text{eff}}=3/2$ states. Could Authors comment on that.

What happens at $|\lambda/t|>2$? Does the flat band remains flat?

4. In Ref. [37] a phase diagram of the Hubbard model for $J_{\text{eff}}=1/2$ states is studied. Why should it be relevant for the Hamiltonian (1) which is supposed to be derived for the E2 states?

How the Hamiltonian (1) differs from a model Hamiltonian for $j_{\text{eff}}=1/2$ states?

5. Authors estimate the trigonal splitting of $J_{\text{eff}}=3/2$ states as $E_1-E_2 \sim 0.2$ eV. On the other hand, the band width of W t_{2g} states obtained from band structure calculations for CsW_2O_6 is ~ 2.5 eV which gives a rough estimate for t_{2g} - t_{2g} hopping of the same order as E_1-E_2 . This suggests that hopping between E_2 and E_1 states cannot be neglected. Moreover, at least at zero SOC all 3 t_{2g} orbitals should be taken into account. How Authors justify the use of the E_2 only model for CsW_2O_6 ?

In the supplementary Authors state that $t \sim 0.06$ eV was evaluated from band structure calculations. I could not find such a value in Ref. [33] in which Ry units are used for band plots.

Some other comments:

6. Eq. (S1) should be corrected as the a_{1g} state corresponds to $3z^2-1$ orbital.

7. What is the parameter k mentioned in the supplementary material?

8. $J_{\text{eff}}^z = \pm 1/2$ states in Eq. (S5) are not eigenstates of $J_{\text{eff}}=1/2$ or $3/2$.

9. Why in the middle panel of Fig. S1(c) E_1 and E_2 states are split even for zero SOC?

- Reply to Referee #1

We thank Referee #1 for highly evaluating our manuscript and recommending it for publication. The comments were very helpful to improve the quality of the manuscript. In the following, we reply to the questions addressed. We modified/added explanations to the corresponding part of the manuscript, which are highlighted.

[Referee #1-1] *"In Eq. (2), the tight-binding Hamiltonian of a bipartite graph should have a chiral symmetry which leads to the flat bands being localized on the zero energy and connected with the other dispersive bands below and above. It means that there must have interference between the pyrochlore and diamond sublattices. Thus, Eq. (3) and (6) cannot be obtained directly from Eq. (2). Is there any approximation? We suggest the authors provide more details about the derivation."*

In the previous manuscript, we may have not properly addressed the role of Eq.(2) (the bipartite Hamiltonian, H_{lg}) in the main text, which confused Referee #1. We introduced Eq.(2) to graphically understand the meaning of T_{OD} using Fig.2a, but *not* for the purpose of deriving Eq.(3).

First of all, Eq.(3) is an identity, which does not require Eq.(2) for proof. If Eq. (3) holds, namely, if there exists T_{OD} of dimension $N_O \times N_D$ with $N_O > N_D$ that fulfills Eq.(3), it proves the existence of flat bands. This is because the shape $N_O > N_D$ of T_{OD} mathematically guarantees that there is a zero-eigenvalue of T_{OD} , and its eigen vectors represent the flat band wave function.

Still, there is an exact relationship between Eqs.(2) and (3). Mathematically, multiplying Eq.(2) twice, namely $(H_{lg})^2$ will give a new block-diagonal matrix. The upper block is $T_{OD}T_{DO}$. Used in Eq.(3).

The confusion caused by Eq.(2) may be because H_{lg} is a bipartite Hamiltonian, which may remind of a Lieb lattice with chiral symmetry and flat band. However, in the present case, Eq.(2) has no physical implications itself. Although Eq.(2) is used in the standard explanation of a line graph theory, we recognize that our explanation may have been *"technical"* as mentioned also by Referee #2.

We clarified the explanation on the whole section "spinor line graph theory"; we deleted Eq.(2) and directly introduced T_{OD} in the main text. The $\lambda \neq 0$ case Eq.(5) and (6) are an extension of the $\lambda = 0$ case in Eq.(3), and the parallel proof holds in our spinor-line graph theory. However, our previous explanation may not have been enough, and we also revise this part by introducing a schematic figure (Fig.2d).

[Referee 1-2] " As studied in Ref.[12], for an *s*-orbital Kagome lattice, the presence of spin-orbit coupling opens a gap between the flat bands and the dispersive bands at Γ point and makes the flat bands be topological. While in FIG. 1c and FIG. 1d (with $\lambda = -2t$), the perfect flat bands seem to degenerate with the dispersive bands at the Γ point. Is the degeneracy accidental or protected by any symmetry? What if λ slightly off the fine-tuning value $-2t$? I would also be interested in the topology of the flatten bands if there opens a gap at the Γ point.

The contact of flat bands with dispersive bands at Γ point is neither an accidental *nor* a typical symmetry-protected band degeneracy. It is called "band touching" and occurs only for flat bands; the origin is clarified in [Bergman, Wu, Balents, PRB **78**, 125104(2008)] as follows; It is proven that the number of linearly independent flat band eigenstates is overcomplete *by order-1*. The flat band assigns these excess states to the dispersive band by enforcing the band touching at Γ point. The reason why the flat-band states become overcomplete and why it touches at Γ point is explained in another very recently published paper [Hwang, *et.al.* Phys. Rev. B **104**, 081104 (2021)], using the symmetry representation. The touching is necessitated by the perfect flatness, combined with a unitary symmetry the system has. Our pyrochlore flat band belongs to this case. When the flat band loses its perfect flatness, the gap can open. We can find this for $\lambda < -2t$, and the system becomes topological insulator at half filling.

We added these explanations together with the additional references (Refs.[40,41,42]).

Since both Referees are interested in how the bands develop, we added a series of band structure of the noninteracting Hamiltonian in new Supplementary D.

We have also made additional revisions mainly in the Supplementary material which were not inquired by the Referees (we also made revisions to the explanation on the related part in the main text). Particularly we added the microscopic evaluation of λ / t and updated the explanation on the details of many-body wave functions. We believe that these revisions improve the accuracy and the quality of the manuscript.

- **Reply to Referee #2**

We thank Referee #2 for valuable comments on the manuscript. His/her insight as an expert of band structural calculation helped us a lot to clarify the presentation of the manuscript.

Particularly, we reexamined the model parameters using the direct comparison of microscopic formulation with the first principles band structural calculations to establish the relevance for CsW₂O₆. We also evaluated t and λ microscopically using the Slater-Koster parameter to verify the realization of our SOC flat bands in the material.

We agree that the first principles calculations are primarily important in understanding the details of the system. At the same time, a minimal model plays a complimentary role to focus on particular properties of the system and clarify the dominant mechanism of the phenomena, as we did in the present work.

To resolve the worries that the spinor line graph theory may be rather technical, we extensively revised the explanation on that part. The mathematics used there itself is elementary and can be understood by undergraduates as well. Although the proof may look technical or specific, it serves as a backbone of our theory, and its basic structure of this proof is quite well known to theorists. We appreciate if Referee #2 considers this point in examining the updated manuscript.

We replied to all questions addressed and revised the manuscript accordingly. We make the point-by-point reply in the following.

[Referee #2-1] "*On p.1 Authors state that the E₂ doublet carries momentum $J_{eff}=3/2$, $J_{eff}^z=\pm 1/2$. However, trigonal crystal field (CF) couples $J_{eff}^z=\pm 1/2$ states with $J_{eff}=1/2$ and $3/2$ and, consequently, E₂ cannot be an eigenstate of J_{eff} .*"

We agree that the E₂ doublet is *not an exact eigenstate* of J_{eff} and J_{eff}^z .

It is an approximate eigenstate of J_{eff} and J_{eff}^z , since we are considering a finite trigonal distortion. However, assigning quantum numbers $J_{eff} = 3/2$, $J_{eff}^z = \pm 1/2$ is convenient to understand the major contribution of angular momentums and types of orbitals to these doublets, which is a quite commonly accepted way of explanation. For example, in Iridates in a relevant paper [PRL **110**. 076402(2013)], the authors perform a RIXS measurement showing that a trigonal splitting has 110meV energy, and mentions that J_{eff} -picture works well, labelling E₃ doublets as $J_{eff} = 1/2$, although is not an exact eigenstate of J_{eff} . There are several other cases considering the similar situation, with larger trigonal distortion angle θ , e.g. Ref.[51] PRB 92, 094405 (2015) in the Supplementary material.

Thanks to the comment, we understand that we need to be careful in clarifying the meaning of this J_{eff} . We revised the explanation and replace $J_{eff} = 3/2$ with $J_{eff}\sim 3/2$, etc. in both the main text and in the Supplementary material.

[Referee #2-2] *"What approximations are made when deriving the Hamiltonian (1)? In particular: Without SOC the E₂ doublet is formed by pure a_{1g} states while with increasing SOC strength it becomes a mixture of a_{1g} and e_g^π states. On the pyrochlore lattice the magnitudes of a_{1g}-a_{1g} and e_g^π-e_g^π hopping matrix elements are very different. Is it taken into account in (1)?"*

The approximation made in Eq.(1) is to take the values of t and λ bond independent uniform constants. Since the form of Hamiltonian operators in Eq.(1) is determined by the site-centered inversion symmetry and time reversal symmetry, it applies for trigonal crystal field. For the perfect octahedral crystal field, t and λ are spatially uniform constants. However, a slight trigonal distortion makes part of the tetrahedron bonds longer and others shorter. In order to see that the results do not change much by approximately taking t uniform in (1), we have shown in Fig. 4 that the trimerized charge order is stable even though we vary t' and t , under such trigonal distortion.

To clarify these points, we add a short comment in the main text on the approximation in Eq.(1) and its implication and relationships with Fig. 4.

As noticed by Referee#2, (see also Supplementary Eq.(S8)), E₂ doublet consists of the linear combination of a_{1g} and e_g^π orbitals, which have comparably large weights when the bare SOC ζ becomes large ($\sin \delta \sim 0.4$, $\cos \delta \sim 0.6$).

We indeed take account of the hoppings between different/same species of orbitals: the $a_{1g} - a_{1g}$ and $e_g^\pi - e_g^\pi$ hopping both contribute to t , whereas $a_{1g} - e_g^\pi$ hoppings contribute to λ . Therefore, increasing SOC (ζ) will reduce t and increase $|\lambda|$ ($\lambda < 0$). We present this quantitatively in the present revision; we performed the microscopic calculation to derive these values using the material parameters in Supplementary A and B, and the Slater-Koster parameters. The actual values of λ/t is found to fall in the range $-3 \sim -1$ for CsW₂O₆ (see Supplementary C and Fig. S3), which is the range where our SOC flat band is reasonably realized.

[Referee #2-3] *"Bands in the $\lambda/t=0$ panel in Fig. 1b look like bands formed by s-like states put on the pyrochlore lattice which is OK for a_{1g} bands. On the other hand, the $\lambda/t=-2$ bands look very similar to $J_{eff}=1/2$ bands which I get from band structure calculations for 5d pyrochlore compounds in the limit of large SOC. Naively, I would expect different dispersion for bands formed by a linear combination of $J_{eff}=1/2$ and $J_{eff}=3/2$ states. Could Authors comment on that. What happens at $|\lambda/t|>2$? Does the flat band remains flat?"*

Indeed, for $\lambda/t = 0$ the band is a a_{1g} -based one. For finite λ/t the E₂ energy band may resemble that of $J_{eff} = 1/2$ of the E₃ doublet, although Referee #2 may not expect them to be.

As we presented in Supplementary Eq.(S8), E_2 doublet ($J_{eff} \sim 3/2$ $J_{eff}^z \sim \pm 1/2$ bands) consists of the linear combination of a_{1g} and e_g^π orbitals with the coefficients $\cos\delta$ and $\sin\delta$. The E_3 doublet ($J_{eff} \sim 1/2$ bands) consists of the same combination of orbitals with the coefficients $-\sin\delta$ and $\cos\delta$. Therefore, it is clear that they share the same Hamiltonian Eq.(1) except that the values of t and λ differ.

When $\lambda/t < -2t$, the flat band no longer remains perfectly flat and also a gap opens at Γ point. Since both Referees are interested in how the bands develop, we added a series of band structure of the noninteracting Hamiltonian in new Supplementary D.

[Referee #2-4] "*In Ref. [37] a phase diagram of the Hubbard model for $J_{eff}=1/2$ states is studied. Why should it be relevant for the Hamiltonian (1) which is supposed to be derived for the E_2 states? How the Hamiltonian (1) differs from a model Hamiltonian for $j_{eff}=1/2$ states?*"

As we replied in [#2-3], the E_3 doublet ($J_{eff} = 1/2$) shares Eq.(1) but with different values of t and λ .

[Referee 2-5] "*Authors estimate the trigonal splitting of $J_{eff}=3/2$ states as $E_1-E_2 \sim 0.2$ eV. On the other hand, the band width of $W t_{2g}$ states obtained from band structure calculations for CsW_2O_6 is ~ 2.5 eV which gives a rough estimate for $t_{2g}-t_{2g}$ hopping of the same order as E_1-E_2 . This suggests that hopping between E_2 and E_1 states cannot be neglected. Moreover, at least at zero SOC all 3 t_{2g} orbitals should be taken into account. How Authors justify the use of the E_2 only model for CsW_2O_6 ?*

In the supplementary Authors state that $t \sim 0.06$ eV was evaluated from band structure calculations. I could not find such a value in Ref. [33] in which Ry units are used for band plots."

Thank you for the comment. First of all, the parameters values we evaluated properly explains CsW_2O_6 . However, we recognized that the explanation of our previous Supplementary A was insufficient. We updated the formulation in Supplementary A in a step-by-step manner. We also carefully reexamined the evaluation of material parameters by comparing the formulation in Supplementary A with the first principles band structure in (newly created) Supplementary B. The conclusion remain the same: $t \sim 0.06$ eV, $E_1 - E_2 \sim 0.2$ eV.

We also added the more reliable estimation about the details of other parameters: $\Delta_1 \sim 0.23$ eV, $10Dq \sim 2$ eV, $a_2 \sim -0.142$.

We reply to the comment using the two figures shown below:

The left panel shows [Fig. 3, right panel in Ref. [33]: Okamoto, et.al. *Nature Comm.* **11**, 3144 (2020)]. The right panel is the first principles band structure by Dr. Youichi Yamakawa (our new

Supplementary B Fig. S2). The l.h.s. one from Ref.[33] was also performed by Dr. Yamakawa (although the authors did not mention it), together with the r.h.s. ones in the same series of calculation.

The l.h.s includes SOC but r.h.s one does not. The comparison of energy levels without SOC gives a clearer evaluation of parameters since the structure is simple, and thus can be directly compared with Eqs.(S1)-(S5) with $\zeta = 0$.

Referee #2 mentions that the bandwidth is read as $\sim 2.5\text{eV}$ from Ref.[33]; The energy window from about 0.5Ry to 0.7Ry indicated in arrows is actually $\sim 2.5\text{eV}$, but includes the contributions from the mixing of all t_{2g} orbitals E_1, E_2 , and E_3 levels as well as the related O-orbitals. We consider a smaller energy window approximately marked with yellow.

In the r.h.s. panel without SOC, a similar window range is marked with yellow. This is the range of a_{1g} bands whose width is $\sim 0.5\text{eV}$. Actually, panel (b) shows that the contribution from $L_z = \pm 1$ (e_g^π) and $L_z = 0$ levels (a_{1g}) is dominant in the pink and yellow ranges, respectively. Since the bandwidth at $\lambda = 0$ or $\zeta = 0$ is $8t$, we obtain $t \sim 0.06\text{eV}$ for the case of $\zeta = 0$.

Introducing ζ , λ will not modify t by orders of magnitude. In fact in the previous manuscript, we compared the band structures of nonzero SOC (which is more complicated, and may not be clear enough) and obtained the same $t \sim 0.06\text{eV}$.

We can also find in the top right panel (a) that there is a mode repulsion between e_g^π and e_g^σ bands (e.g. blue and red lines); the center of these two levels can be accurately determined as the

broken line at around 1.4eV. The a_{1g} level is also determined at around 0.2eV by the nondispersive R-M line (see also Fig.1b in the main text with $\lambda=0$). Their energy difference is analytically given by Eq.(S5), and from this comparison we can determine $a_2 \sim -0.142$. (a_4 is determined by $\theta=55.71^\circ$). The evaluation $10Dq \sim 2$ eV is also given reasonably from the same energy band. Therefore, the material parameters except for ζ are fully determined by the above comparison. By introducing $\zeta=0.1$ ($10Dq$) to the Eq.(S7), we obtain $E_1 - E_2 \sim 0.2$ eV.

These parameter values are consistent in their order of magnitudes from the parameters of other 5d materials which we newly listed in Table I in Supplementary A.

We disagree with part of the comment saying that “*at least at zero SOC all 3 t2g orbitals should be taken into account*”. This is because we are *not* targeting the $\zeta=0$ case, and Eq.(1) does not necessarily need to explain this situation. We are not arguing that the zero SOC of the model ($\lambda=0$) explains some unknown particular pyrochlore material without SOC ($\zeta=0$). The case with large $\zeta > 0$ realized in CsW₂O₆ is related to large $\lambda/t < -1$ region of Eq.(1) (Supplementary C).

At the same time, to understand what happens to the model Eq.(1) when the parameter λ/t is varied is a purely theoretical issue, which we suppose, match the interest of Referee#2.

To summarize this part, the material parameters are safely evaluated with minimal assumption, and the splitting of E_1, E_2 is reasonably large compared to t and the two energy levels seem to be quite well separated as anticipated from the first principles band structural calculation with SOC. We also notice that there is a nearly flat energy band near the Fermi level in the l.h.s. panel with SOC (the dispersive branch below has a large contribution from the O-orbitals and other orbitals, as we can see in panel(b)).

Therefore, it is reasonable to consider that Eq.(1) serves as a minimal model of CsW₂O₆.

[Referee 2-6] "*Eq. (S1) should be corrected as the a_{1g} state corresponds to $3z^2-1$ orbital.*"

We apologize for typo's, and thank Referee#2 for pointing out.

[Referee 2-7] "*What is the parameter k mentioned in the supplementary material?*"

The parameter κ represents the ratio of coefficients of spherical harmonics used to expand the crystal field Hamiltonian. We added detailed explanation in Eq.(S1)-(S3) and the details of the evaluation of $\kappa = \kappa_{sl}$ by the Slater's rule is given in the last part of page 1 in Supplementary A.

However, during this evaluation, we noticed that the value of κ_{sl} does not properly reproduce the experimental observation nor the band structural calculation of 5d materials listed in Table I.

This is because the a_2 term which includes κ is largely modified by the effect from other metallic ions (while a_4 and b do not), which are not taken account in Eqs.(S1)-(S5). In Ref.[35] the authors introduce the other terms that increases the amplitude of negative a_2 by orders of

magnitude for the particular crystal. We thus made a_2 an independent parameter and determined it from the comparison with the band structure as we explained in the previous page of this reply.

[Referee #2-8] " $J_{\text{eff}}^z = \mp pm$ 1/2 states in Eq. (S5) are not eigenstates of $J_{\text{eff}} = 1/2$ or $3/2$."

Please find the reply we made for [Referee #2-1] of the same content.

[Referee 2-9] "*Why in the middle panel of Fig. S1(c) E_1 and E_2 states are split even for zero SOC?*"

We apologize for the mistake, which have confused Referee#2. In the previous Supplementary we have made several typo's and trivial mistakes in the calculation. We corrected it. We now find that when ζ is zero, $E_1 = E_3$ (see Fig. S1(c)).

List of changes:

The changes made in the main text (except for the trivial modification of definitions and indices, English expressions) are highlighted. Supplementary A is fully updated (not highlighted.) Supplementary B, C, D are added, and G is modified together with the related part in the main text. We also would like to slightly change the title since the chirality and charge orderings are independent phenomena although the both appear in the present model.

- Page 1, Model system: explanation on the effective J_{eff} is added, and the Supplementary A & B is noticed. [#2-1]
- Page 2: The explanation on Eq.(1) is added. [#2-2]
- Page 3, at the end of Model system: One paragraph explaining the "band touching" is added. [#1-2]
- Page 4: Explanation on the incidence matrix is updated. Explanation on the spinor-line graph theory is improved with an additional figure (Fig.2d). [#1-1]
- Page 6: Discussion on the material parameters are modified [#2-2,3,4,5]
- Supplementary A: rewritten and the figures are updated. [#2-5,6,7,8,9]
- Supplementary B: newly created [#2-3]
- Supplementary C: newly created, evaluating λ/t microscopically. [#2-2]

- Supplementary D: newly created [#1-2, #2-3]
- Supplementary G: the explanations on the wave function is updated.
- Page.6, The explanation of the form of many-body wave function in the main text is modified.
- We corrected typo's and revised the English expressions throughout the paper to make it more easier to read.
- Refs.[40,41,42] added: [#1-2]

Regarding Supplementary G, we note that during the modification, we recognized that the number of independent one-body flat-band states are restricted by symmetry of the lattice, and the previous description on the many-body wave function was rather misleading. We modified that part in the main text(highlighted), and the related Supplementary G.

We also corrected several notations for the clarification of the explanation. The nontrivial ones are highlighted, but does not change the formulation nor the conclusions. (e.g. Sign of 2nd term of Eq.(5) changed due to the change of definition of the sign of variables. Making Eq.(6) a vector instead of a scalar.)

Since the information on the band structure was not available for some time which we made use of in this revision in Supplementary B, the revision took more time than we first expected. We apologize for the delay.

REVIEWERS' COMMENTS

Reviewer #1 (Remarks to the Author):

I would like to thank the authors for their answers to my questions.

The paper has been improved. I do think that the paper can be published.

Reviewer #2 (Remarks to the Author):

Authors provided detailed replies to all my comments and did appropriate changes to the manuscript. The paper can now be published in Nature Communications.